# A variational autoencoder trained with priors from canonical pathways increases the interpretability of transcriptome data

**Bin Liu**[1], **Bodo Rosenhahn**[2], **Thomas Illig**[1,3], **David S. DeLuca**[1] *

**1** Hannover Medical School, Biomedical Research in Endstage and Obstructive Lung Disease Hannover (BREATH), German Center for Lung Research, Hannover, Lower Saxony, Germany, **2** Institut für Informationsverarbeitung (TNT), Leibniz University Hannover, Hannover, Lower Saxony, Germany, **3** Hannover Unified Biobank, Hannover Medical School, Hannover, Lower Saxony, Germany

* DeLuca.David@mh-hannover.de

## Abstract

Interpreting transcriptome data is an important yet challenging aspect of bioinformatic analysis. While gene set enrichment analysis is a standard tool for interpreting regulatory changes, we utilize deep learning techniques, specifically autoencoder architectures, to learn latent variables that drive transcriptome signals. We investigate whether simple, variational autoencoder (VAE), and beta-weighted VAE are capable of learning reduced representations of transcriptomes that retain critical biological information. We propose a novel VAE that utilizes priors from biological data to direct the network to learn a representation of the transcriptome that is based on understandable biological concepts. After benchmarking five different autoencoder architectures, we found that each succeeded in reducing the transcriptomes to 50 latent dimensions, which captured enough variation for accurate reconstruction. The simple, fully connected autoencoder, performs best across the benchmarks, but lacks the characteristic of having directly interpretable latent dimensions. The beta-weighted, prior-informed VAE implementation is able to solve the benchmarking tasks, and provide semantically accurate latent features equating to biological pathways. This study opens a new direction for differential pathway analysis in transcriptomics with increased transparency and interpretability.

## Author summary

The ability to measure the human transcriptome has been a critical tool in studying health and disease. However, transcriptome datasets are too large and complex for direct human interpretation. Deep learning techniques such as autoencoders are capable of distilling high-level features from complex data. However, even if deep learning models find patterns, these patterns are not necessarily represented in a way that humans can easily understand. By bringing in the prior knowledge of biological pathways, we have trained the model to "speak the language" of the biologist, and represent complex transcrtomes, in simpler concepts that are already familiar to biologists. We can then apply the tool to

**Data Availability Statement:** The source code for this paper is available on GitHub with the following link: https://github.com/BinLiu9205/deepRNA_autoencoder.git and on figshare with the following

link: https://figshare.com/articles/software/
deepRNA_autoencoder/22227217.

**Funding:** This research was supported by the
German Center for Lung Research (DZL, https://
dzl.de/), grant number 82DZL002A1, via the
Federal Ministry of Education and Research
(BMBF), which provided salaries for BL and DSD.
The funders had no role in study design, data
collection and analysis, decision to publish, or
preparation of the manuscript.

**Competing interests:** The authors have declared
that no competing interests exist.

compare for example samples from lung cancer cells to healthy cells, and show which biological processes are perturbed.

## Introduction

Transcriptomics is a powerful tool in characterizing cellular activity under various conditions, which allows researchers to discover the underlying associations between transcripts or genes and pathological or environmental factors. Therefore, transcriptomics data are widely applicable in multiple areas of biomedical research, varying from understanding disease mechanisms [1], detecting biomarkers [2, 3], to tissue-specific regulatory gene identification [4]. The broad application of the technology leads to generating a considerable amount of transcriptomic sequencing data and constructing a few specific public platforms hosting the relevant biological datasets, such as ArrayExpress [5] and NCBI GEO [6]. In all these biomedical tasks, human-understandable interpretation of the experimental transcriptomics data serves as the pivotal component to understanding the underlying biology. However, this interpretation remains a challenge in the face of large and complex datasets. Here we explore the potential for machine learning models to learn a simplified representation of the transcriptome in terms of commonly understood biological processes, and thus increase the interpretability.

While gene set enrichment analysis (GSEA) [7] is a standard tool for interpreting regulatory changes to the transcriptome, the method highly relies on a list of well-detected differentially expressed genes (DEGs) between the conditions of interest. Furthermore, most of the state-of-the-art models for differential expression analysis (DEA) are based on the linear assumption across samples, the representatives of which include the models from limma [8, 9], DESeq2 [10], and Seurat [11]. However, variation in measured expression levels, whether of biological or technical origin, may not always behave linearly. Given the potential for synergistic effects between genes, this assumption of linearity might lead to a loss of power. Complex sources of variation are also present in combined public datasets, consisting of a large number of samples from multiple sources, and influenced by non-biological factors such as batches or processing centers. These limitations daunt the exploration of large-scale transcriptomics data for answering fundamental biological questions.

The goal of this study is to utilize deep learning techniques to bring transparency into which biological process patterns are represented in a transcriptome dataset, and thus to facilitate the interpretation of experimental results. Deep learning with artificial neural networks has witnessed rapid development in recent years and outperformed the traditional approaches in handling massive and complex data from multiple areas because of its high-level feature extraction at a non-linear space and objective data processing [12–16]. This development also enables more possibilities in understanding the transcriptomics data and has successfully contributed, for example to drug repurposing and development [17, 18], phenotype classification [19] and genomics functional characterization [20] with a more in-depth understanding of transcriptomics data.

We are specifically interested in autoencoders as a class of methods that can reduce dimensionality and learn the major features in complex data [21]. Autoencoders achieve this by passing the data through a bottleneck layer (a layer with fewer nodes than in the input layer) and optimizing the model with the objective of generating an output that is as similar as possible to the input. These methods have been explored in the context of transcriptomes in a few representative publications, including [20, 22–26], demonstrating that interesting biological features are captured in the latent space of the model. However, these studies take different

approaches to the challenge of associating latent representations with human-understandable biological concepts. [22] implemented an autoencoder and interpreted that latent space by correlating latent features with phenotypes post hoc. [20] on the other hand sought to constrain the latent space to represent known pathways by restricting the network connectivity, i.e., each latent node represents a pathway, and only genes known to be involved in that pathway are connected to the node upstream. In the study of [22], the network is free to learn any representation from that data, but the burden of interpretation is left to post hoc analysis. In the case of [20], the network is restricted directly in its architecture based on gene set definitions.

Here, we see an opportunity to implement a solution that finds a middle path in which the network is encouraged to learn a latent representation based on known biological concepts, but still has the freedom to learn relationships among genes from the data. Specifically, we propose an autoencoder variant using a novel technique of introducing pathway-informed priors. The basis for this approach is the Bayesian framework implemented in variational autoencoders (VAE) [27]. The VAE framework inherently provides the opportunity to involve priors in the training process to learn latent representations. With the introduction of biologically meaningful priors, our approach here aims to integrate prior knowledge from the domain (here, we use Hallmark pathways defined by MSigDB and selected pathways as two examples) and still retain the flexibility of the data-driven deep learning approach. This approach is most comparable to those presented by Zhao [25] and Lotfollahi [26], with the commonality that the goal is to produce latent features that correspond to known biological concepts. However, compared to our approach of incorporating canonical pathway knowledge as priors, both Zhao [25] and Lotfollahi [26] provide pathway definitions to constrain the decoder architecture. The prior-based approach provides an additional opportunity to calibrate the strength of the effect of prior following the established beta-VAE approach [28].

Specifically, We make use of a hyperparameter, beta, in a similar way described by [28], which can add weight to the influence of the priors on the training solution. Thus, using the beta, we can control the extent to which the model conforms to pathway concepts previously defined by MSigDB or KEGG versus being free to define latent variables in any way that best encodes the transcriptome in a reduced representation. The fine-tuning of this hyperparameter enables control over the tension between direct biological interpretability and the ability to deviate from canon or find new patterns.

In this study, we implement and compare several standard autoencoder implementations: (i) fully connected autoencoder (simpleAE) [29], (ii) variational autoencoder (simpleVAE), (iii) beta-VAE (beta-simpleVAE), as well as (iv) novel derivatives using prior biological data: priorVAE, and (v) beta-priorVAE. In order to benchmark the performance of the series of models proposed here, we perform tissue and disease classification (e.g., adenocarcinoma, small cell lung cancer (SCLC), leukemia) based on the latent variables discovered in the models. This study explores the feasibility of the prior-based VAE approach in increasing the transparency and interpretability of transcriptomes, as well as how the hyperparameter beta controls the balance of using prior information versus learning novel patterns directly from the data.

## Materials and methods

### Datasets and preprocessing

The dataset employed for the training of the model and subsequent analyses was downloaded from ArrayExpress [5], with Accession ID E-MTAB-3732 [30]. This dataset comprises 27,887 Affymetrix HG-U133Plus2 arrays, sourced publicly. All samples underwent quality control filtering and were annotated for disease status and cell line information. The data was

normalized using fRMA [31–34] within the R Bioconductor platform by the dataset's author. The dataset contains samples from healthy individuals, those with diseases (including cancer), and cell lines. The original dataset has been divided into a training and a test set at an 8:2 ratio, with stratification based on the source organs of the samples. The curated gene signature dataset, used in the previous generation, is the Hallmark gene set, sourced from Human MSigDB Collections [35]. We also sourced pathways from KEGG [36–38], which were filtered to remove disease pathways.

The human transcriptomes in this dataset contained 54,675 transcripts per sample. Because of concerns around overfitting and model performance for such a large input layer, efforts were made to reduce the representation of the transcriptome prior to passing it as input into the neural network. Thus, the experiments included variations in input type. In the first input variation, the data were fed into the model on the transcriptome level without further processing. For the second input variation, the transcripts were collapsed into the gene-level using the platform annotation offered by Gene Expression Omnibus (GEO) [6]. The normalized expression values of the transcripts were transformed into the original level by applying a power function. The original expressions were averaged, and the original scale was restored with a log2 transformation. This resulted in an input size of 23,375 genes.

For the third input variation, the goal was to decrease the number of trainable parameters in the models further. We performed a community detection, and then selected a gene for each community as a representative of the whole community. The community detection algorithm began with gene-level expression values. A graph representation of the set of genes was then initialized with a k-nearest neighbors approach (NearestNeighbors function from the sklearn Python package [39]), with edge weights defined as the absolute value of the correlation between the two genes across samples. The next step was to detect clusters in the gene graph. We use the Leiden algorithm [40] to define clusters, which we interpret as communities in this graph. A total of 2032 communities were defined using a resolution value of 0.02. A single gene was chosen as a representative of each community for use as the final input. Community representatives were based on the criteria of having the highest sum of correlation with all the other genes in the same community.

## Model architectures

The experiment systematically trained and compared five architectures of autoencoders, including one fully-connected autoencoder without prior information (simpleAE) and four variational autoencoders (VAE) architectures. Besides the variational autoencoder with unit Gaussian prior (simpleVAE) and the beta-constrained variational autoencoder with unit Gaussian prior (beta-simpleVAE), we also presented a novel technique of introducing pathway-informed priors (priorVAE) and tested the influence of the hyperparameter beta over this biological relevant prior VAE (beta-priorVAE).

We construct three fully-connected linear layers in the encoder of all the autoencoder architectures (number of trainable parameters equal to the dimension of features, 1000 and 100, respectively). The leaky rectified linear unit (Leaky ReLU) serves as the activation function between encoder layers.

The bottleneck layer of the simpleAE is a dense layer with 50 dimensions. For the VAEs, the bottleneck is implemented as two fully-connected, 50-dimension layers: one for learning $\sigma$s and one for learning $\mu$s, which are then sampled using the reparametrization trick before going on to the decoder. The decoders in all models mirror the three dense layers of the encoders. The Softplus activation function is added to the last layer of the decoder for reconstruction to output values on the same scale as gene expression input values.

The loss function for the simpleAE is the mean squared error (MSE) between the decoder output and the original input values. For VAE, the loss function has two terms: (i) the reconstruction loss, also MSE, and (ii) the KL divergence between the latent distributions and unit Gaussian prior distributions. For the prior- and beta-priorVAE implementations, the loss functions are described in detail below.

## The preparation of pathway-informed priors

Pathway-informed prior distributions were generated for each sample in the form of $(\sigma^2)$ and $(\mu)$ parameters using a bootstrapping procedure. We defined $\mu$ as the average gene expression level of genes within each pathway definition, and $\sigma^2$ as the variance across bootstrapping iterations. Pathway definitions were taken from MSigDB Hallmarks [35] or the KEGG database [36–38].

## Loss function for biological priors

Generally, variational autoencoders take the form found in Fig 1 and have been previously described in detail [27]. In short, it has been established that neural network training can perform variational inference when the loss function takes the form:

$$= \mathbb{E}_{q_\phi(z|x)}[log p_\theta(x|z)] - KL(q_\phi(z|x)||p(z)) = L_{recon} + L_{KL} = L_{VAE architectures}$$

Here, $x \in \mathbb{R}$ is a vector of expression values for a sample in the set of all samples, $X$. The generative model $p_\theta(x|z)$ is learned, where $z$ ($z \in \mathbb{R}$) are latent variables with a prior $p(z)$ such that $z$ can generate the observed data $x$.

The reconstruction loss term $L_{recon}$ can be measured by the mean squared error (MSE) between the input and the reconstructed output. In most VAE implementations, a Gaussian distribution is used for $p(z)$ to make a tractable KL calculation. The assumption of unit Gaussian, $\mathcal{N}(0, I)$, as priors leads to the simplified expression:

$$KL(q_\phi(z|x), p(z)) = \frac{1}{2}[-(\log \sigma^2 + 1) + \sigma^2 + \mu^2]$$

In order to implement pathway-informed priors with any parameter values, the KL divergence had to be implemented more generically for any two Gaussian distributions to measure the distance between $q_\phi(z|x)$ and $p(z)$, where $q_\phi(z|x) \sim \mathcal{N}(\mu_1, \sigma_1^2)$ and $p(z) \sim \mathcal{N}(\mu_2, \sigma_2^2)$:

$$KL(q_\phi(z|x), p(z))$$

$$= -\int p(z) \log q_\phi(z|x) dx + \int p(z) \log p(x) dx$$

$$= \log \frac{\sigma_2}{\sigma_1} + \frac{\sigma_1^2 + (\mu_1 - \mu_2)^2}{2\sigma_2^2} - \frac{1}{2}$$

The models beta-simpleVAE and beta-priorVAE involve the addition of hyperparameter $\beta$ ($\beta > 1$) to put more weight on the $L_{KL}$ term, as described in [28]:

$$L_{beta-VAE architectures}$$

$$= \mathbb{E}_{q_\phi(z|x)}[\log p_\theta(x|z)] - \beta KL(q_\phi(z|x)||p(z))$$

$$= L_{recon} + \beta L_{KL}$$

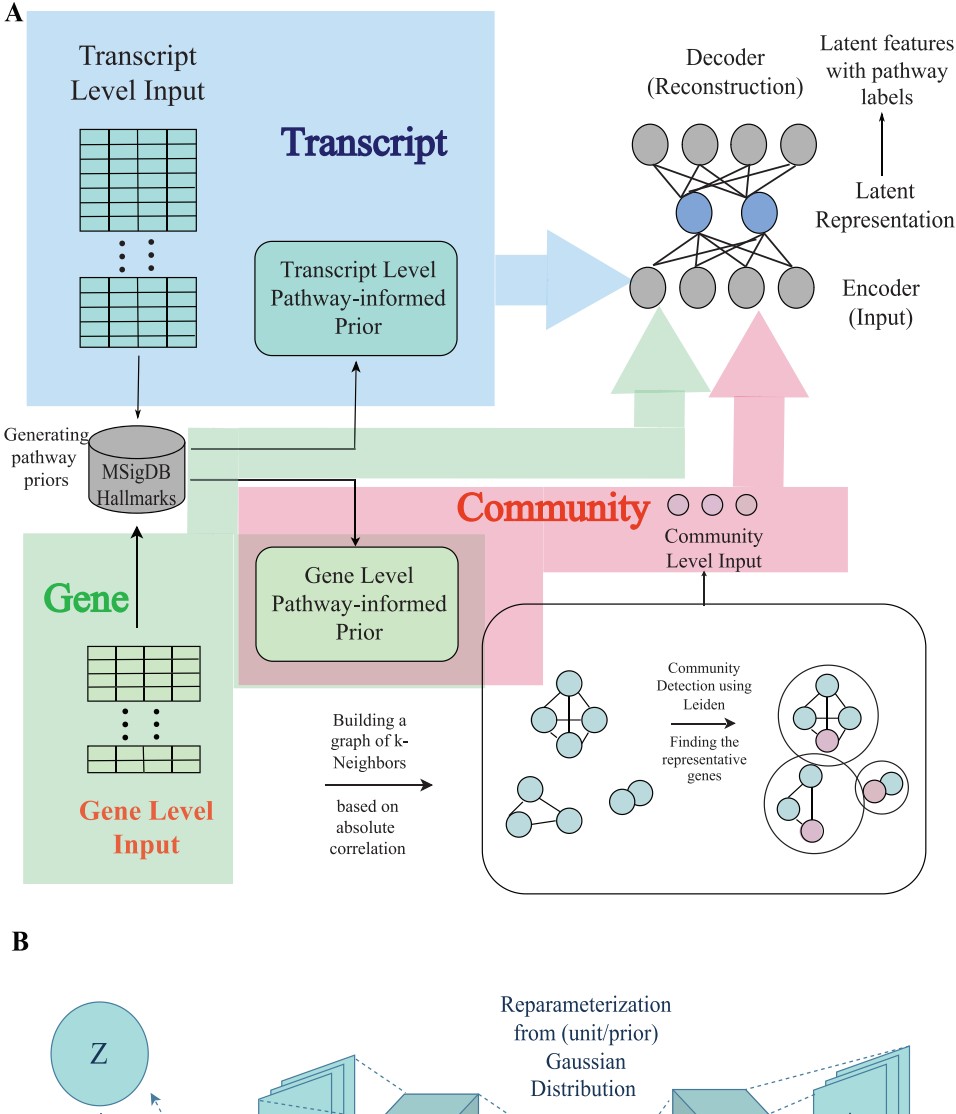

**Fig 1. A conceptual illustration of variations autoencoder architectures.** A: An overview of the system architecture. A pathway-derived prior is generated on the transcript or gene level, as described in the following sections. The three alternative input types (transcript-level, gene-level, and community-level) correspond to three different model variants. B: As a latent variable framework, the model assumes that latent variables ($Z$) are determinants of the measured data ($X$). To learn $Z$, $p(Z|X)$ is approximated by $q(Z|X)$, and modeled as the encoder portion of the network. The latent values represent probability distributions, which are implemented in the bottle-neck layer as values for mu ($\mu$) and sigma ($\sigma$). Finally, the decoder is conceptually equivalent to $p(X|Z)$.

### Evaluation methods

**Hyperparameter analysis.**   The hyperparameter analysis was conducted using the sweep function from the wandb platform. The hyperparameter search space consisted of variables for model architectures, beta, learning rate, batch sizes and epoch. Using multidimensional random sampling, 54 value combinations were selected. Additional models were generated for systematically walking over values for beta.

**Classification benchmarks.**   Five datasets were selected as classification tasks for the purpose of benchmarking. Classifiers utilizing latent dimensions of the models being tested as features were implemented using the LogisticRegression function (with the solver set to 'lbfgs') from the sklearn Python package [39]. The performance score was determined using the cross_validate function from the same package. For the multiclassification task of tissue identification, the analysis was performed iteratively on healthy samples from the eight most prevalent organs. The classification task was framed as distinguishing one target tissue from all other tissues. The average score was computed across all combinations.

**Differential gene expression and gene set overrepresentation analysis.**   The differential expression analysis was conducted using limma [9] and corrected for batch effects by considering the study from which the sample originated as a batch. Significantly differential expressed genes are defined as those having adjusted p-values smaller than 0.05. The significant genes were selected to run the overrepresentation analysis using the EnrichR [41, 42] package in R.

## Results

### Input layer

The Affymetrix HG-U133Plus2 chip captures expression at the transcript level, providing an input space of roughly 50,000 features. Concerned that this would result in a parameter space that was too large for the available data, we experimented with collapsing the inputs to the gene level and even further to 'representative genes' using a network-based clustering technique. A comparison of the choice of inputs can be found in S1 Fig, which reports the correlations between input and output transcriptomes. We concluded that gene-level input was a sufficient reduction in the parameter space based on the results. Furthermore, we explored an alternative gene expression normalization strategy based on z-score, which would have emphasized relative changes in gene expression over absolute changes. However, the standard RMA normalization provided in the original dataset was superior. The correlation between input and their pairwise reconstructed samples has decreased from 0.9852 ± 0.0089 to 0.9007 ± 0.0105 for the priorVAE, and from 0.9691 ± 0.0143 to 0.8587 ± 0.0237 for the beta-priorVAE after training on the z-score normalized dataset with the same model architectures. Due to these initial assessments, the results reported below are based on gene-level input, with the original normalization.

### Learning latent representations with several autoencoder architectures

Five autoencoder variations were trained on the full set of transcriptomes provided by the ArrayExpression dataset, E-MTAB-3732, containing 27887 samples. As shown in Table 1, common to each architecture is a 50-dimensional latent space. For the models, simpleAE, simpleVAE, and beta-simpleVAE the 50 latent dimensions were learned strictly from the data, without attributing any prior biological concepts, and are simply enumerated 1 through 50. For the priorVAE and beta-priorVAE, the 50 latent nodes are associated with the 50 pathways found in the MSigDB Hallmarks gene sets, and accordingly, each latent node can be labeled with the gene set name.

**Table 1. An overview of the structures of the five architectures.**

| Model | Latent Nodes | Beta | Prior | Loss Function |
|---|---|---|---|---|
| SimpleAE | 50 values, unlabeled | none | none | $MSE$ |
| SimpleVAE | 50 Gaussians, unlabeled | none | unit Gaussian | $MSE + \frac{1}{2}[-(\log \sigma^2 + 1) + \sigma^2 + \mu^2]$ |
| Beta-SimpleVAE | 50 Gaussians, unlabeled | beta = 250 | unit Gaussian | $MSE + \beta * (\frac{1}{2}[-(\log \sigma^2 + 1) + \sigma^2 + \mu^2])$ |
| PriorVAE | 50 Gaussians, pathway-derived | none | pathway names | $MSE + \left(\log \frac{\sigma_2}{\sigma_1} + \frac{\sigma_1^2 + (\mu_1 - \mu_2)^2}{2\sigma_2^2} - \frac{1}{2}\right)$ |
| Beta-PriorVAE | 50 Gaussians, pathway-derived | beta = 250 | pathway names | $MSE + \beta * \left(\log \frac{\sigma_2}{\sigma_1} + \frac{\sigma_1^2 + (\mu_1 - \mu_2)^2}{2\sigma_2^2} - \frac{1}{2}\right)$ |

The first step in our evaluation strategy focused on the performance according to the main components of the loss function, as seen in Table 2. (i) reconstruction loss, (ii) the Kullback–Leibler (KL) divergence between the latent distributions and the prior distributions, and (iii) the combined total loss.

Taking these metrics together, it is clear that the introduction of hyperparameter $\beta$ results in a lower KL divergence at the cost of a slightly worse reconstruction loss. The introduction of priors resulted in little change to the loss terms compared to their non-prior counterparts.

To further explore reconstruction performance, we computed correlation coefficients between input transcriptomes vs. the output transcriptomes. A complete pairwise correlation analysis across samples shows how similar input and output transcriptomes are to each other in the context of the natural variability across samples (Fig 2). Each model was able to reproduce reasonable output transcriptomes, which correlated with R between *0.97 and 0.8*. In most cases, the closest pairwise correlations were between the input and output models, although this was not the case for many samples in the beta-simpleVAE, and a single sample in the beta-priorVAE. Together with the performance shown in Fig 2, it is clear that increasing the beta hyperparameter and emphasizing the influence of the priors comes at a cost to reconstruction performance.

We performed further analysis of the reconstruction results to identify whether certain cell types or particular genes were more or less problematic for reconstruction. S2 Fig shows the reconstruction correlations across tissues, with the reconstruction performance being compatible across tissues, despite noticeable heterogeneity in blood samples in particular. To assess whether particular genes were "harder" to reconstruct, we calculated a gene-by-gene error metric and assessed that as a function of expression level (S3 Fig). The gene with the highest error was RPS4Y1, followed by XIST. A further assessment of the reconstruction effects was visualized using t-SNE [43] (S4 Fig). The reconstructed plots indicate similar clustering patterns compared with the original.

**Table 2. Benchmark results for the five architectures on the test set.**

| Model | Recon. Loss | KL(w.$\beta$) | KL(n.$\beta$) | Total Loss (w.$\beta$) |
|---|---|---|---|---|
| SimpleAE | 2674.6 | NA | NA | NA |
| SimpleVAE | 2780.8 | 184.5 | 184.5 | 2965.3 |
| Beta-SimpleVAE | 5899.7 | 7.4 | 1846.8 | 7746.5 |
| PriorVAE | 2758.5 | 160.3 | 160.3 | 2918.8 |
| Beta-PriorVAE | 5740.2 | 8.1 | 2012.9 | 7753.1 |

Loss function values broken down into reconstruction loss, KL divergences with $\beta$ (w.$\beta$), without $\beta$ (n.$\beta$)

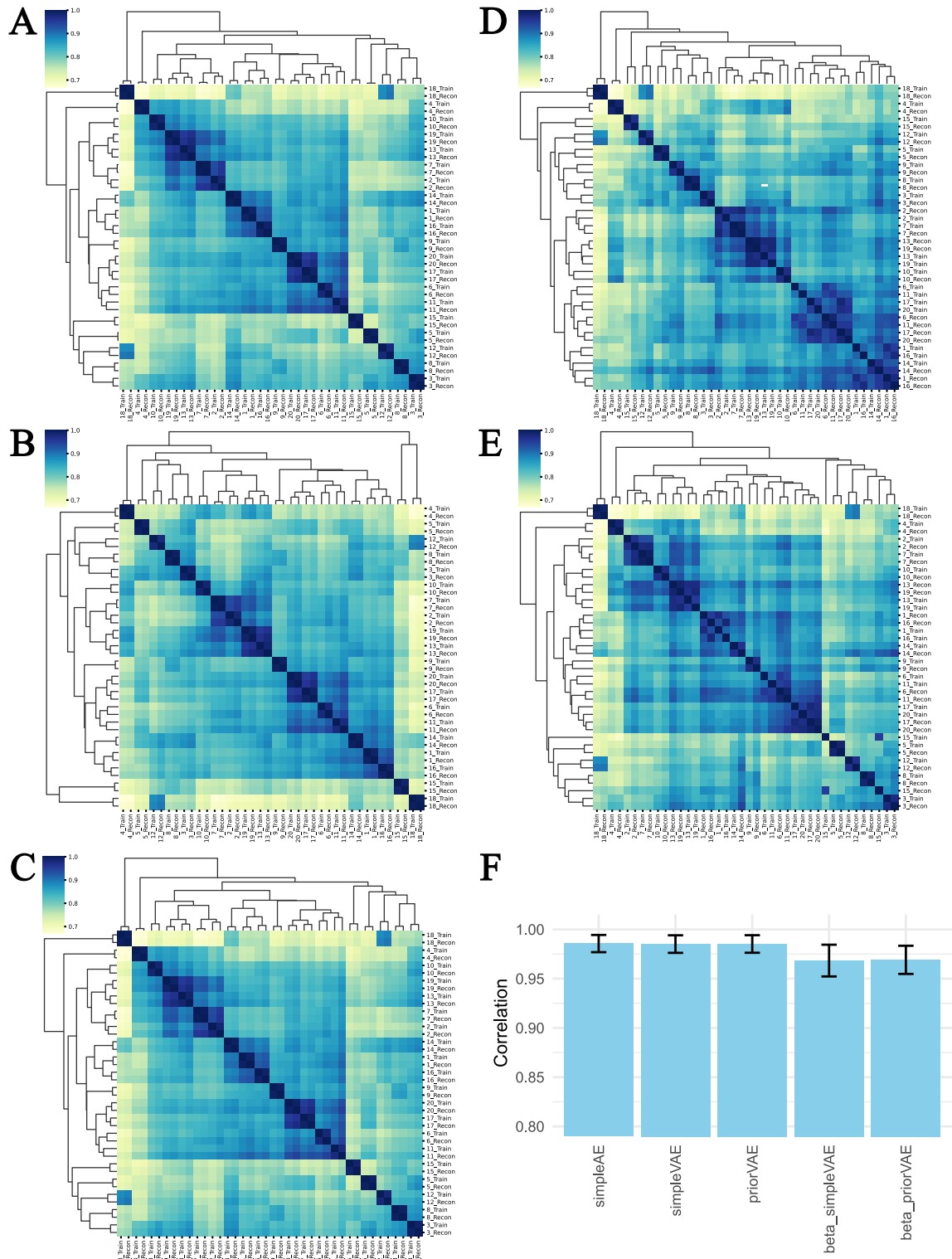

**Fig 2. Reconstruction performances using correlation coefficients between input and output transcriptomes.** A-E: The clustered pair-wise correlation heatmaps of the selected input and their reconstructed output for A: simpleAE, B: simpleVAE, C: priorVAE, D: beta-simpleVAE, E: and beta-priorVAE. Selected input samples and their corresponding reconstruction output are enumerated as 1–20. '_Train' represents the input train sample and '_Recon' represents the reconstructed output. F: The average correlation between the input and its corresponding reconstruction output.

Ideally, the latent dimension should contain enough information to capture the essential biological features. To systematically evaluate how well the models can perform in this regard, we have used the latent features as input into classification models for five validation tasks: (i) distinguishing leukemia vs normal bone marrow, (ii) distinguishing lung vs breast adenocarcinoma, (iii) lung adenocarcinoma vs healthy lung, (iv) lung adenocarcinoma vs small cell lung cancer, and (v) distinguishing tissue types. Of these five tasks, two were too trivial to provide a proper point of comparison among the models: lung adenocarcinoma vs healthy, and lung adenocarcinoma vs small cell lung cancer. However, these comparisons could still be used later in our evaluation of differential pathway analysis. The classification performance after five-fold cross-validation is reported in Fig 3.

Across the three classification models, a clear pattern emerges that the beta-simpleVAE performs worse than the others. The beta-priorVAE also has a tendency to perform slightly lower than the remaining models. For leukemia vs normal, a high precision and slightly lower recall indicates that some leukemia samples are difficult to distinguish from normal. In the case of breast cancer vs lung cancer, the situation is more balanced.

For the benchmarking task of classifying tissue types, eight target organs were selected based on available samples. In this classification task, the simpleAE, simpleVAE and priorVAE perform with an average precision above 80%. Beta-weighting leads to worse classification performance in both beta-simpleVAE and beta-priorVAE. However, beta-priorVAE performs much better in this validation task than the former (roughly 0.76 vs. less than 0.6, Fig 3).

## Impact of hyperparameter choices on performance

When designing the main models presented here, decisions about the layer size, learning rate, batch size, and beta values were made best on reasonable expectations. However, it is important to assess the stability of the model and the consequences that these choices as well. This type of post-hoc hyperparameter analysis has an advantage over a hyperparameter optimization strategy (i.e. searching the hyperparameter space for the best performance and then presenting the best models) because it does not require a third set of data for validation and decreases the risk of over-fitting. The results of this assessment are found in S5B Fig.

Upon performing a multidimensional hyperparameter sweep, the beta coefficient in the loss function had a substantially higher importance score compared to learning rate and batch size (S5A Fig). The impact of the number and size of encoder/decoder layers was relatively modest, with the lowest-loss models found among the two hidden layer models over the single and triple hidden layer variants (S5B Fig).

We performed additional benchmarks as part of a sweep across beta values. The correlation between the latent values and priors increased with beta, tangentially approaching 1.0, with notable saturation around 250, with an average correlation of 0.797 (S5C Fig). Beta was also positively correlated with reconstruction loss and negatively correlated with KL loss (S5D Fig). Across biological classification tasks, betas was generally negatively correlated with performance. Several tests, particularly in the lung cancer classification set, exhibited a drop in performance shoulder at beta = 500 (S5E–S5G Fig).

## Larger latent spaces—KEGG and wild cards

We also investigated the outcome of the modeling with larger latent spaces: 1) by choosing a selection of KEGG pathways [36–38] to represent the latent layer for the prior-based autoencoders and 2) by providing an additional unlabeled latent variable with a unit Gaussian prior. The MSigDB Hallmarks gene set catalog was chosen for models presented above because it is small (only 50 gene sets, facilitating direct human interpretation) and as the name indicates,

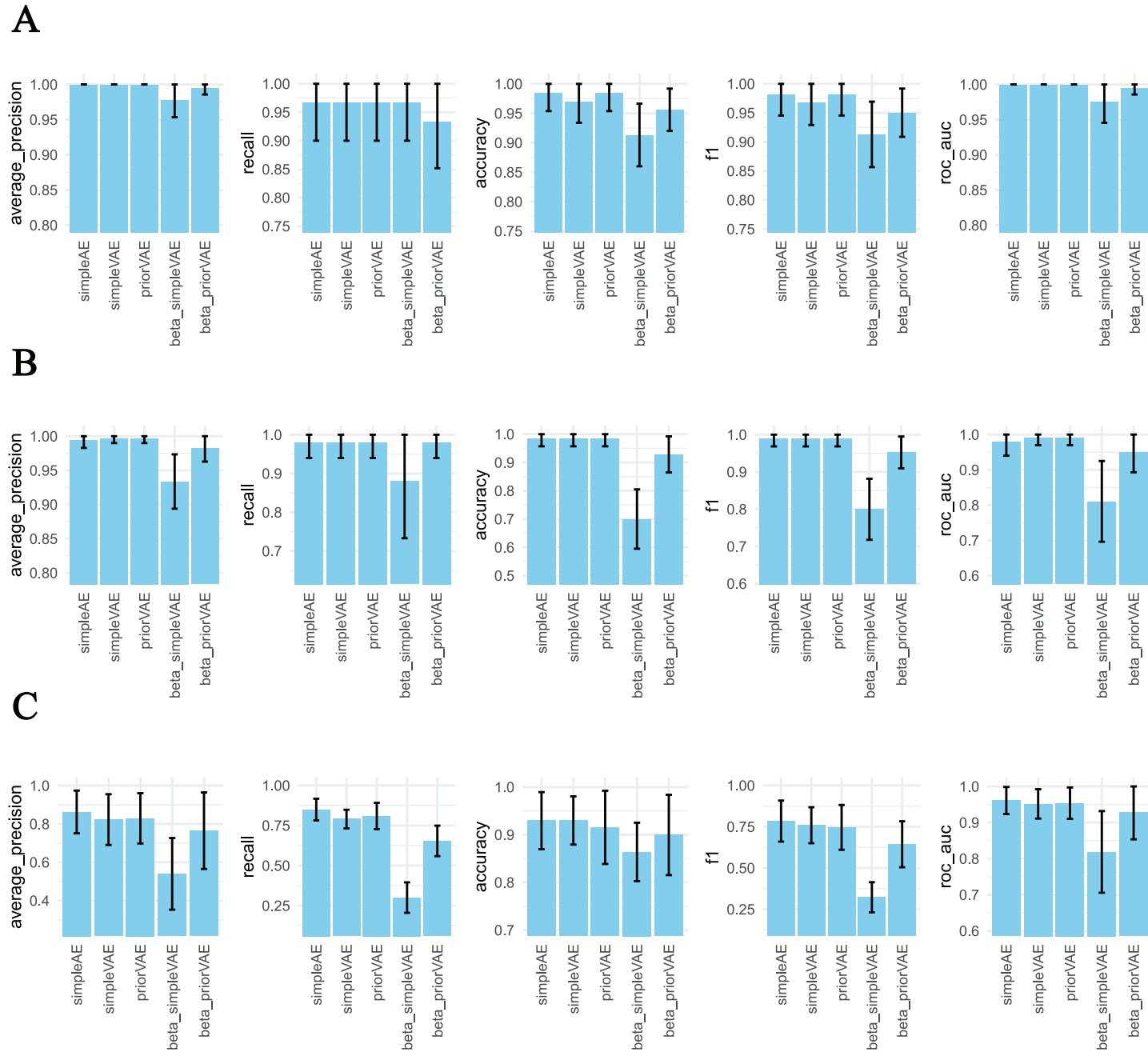

**Fig 3. The performance of the AE models across several sample classification tasks.** Sample classification was based on multivariate logistic regression models as a function of the latent representation provided by each of the autoencoder architectures.

promises to contain the most representative pathways and processes. KEGG is an excellent alternative due to its expert curation, and manageable size. After removing disease-related gene sets, we proceeded with 149 KEGG pathways.

Fig 4 reports the classification benchmarks for models with KEGG vs MSigDB latent spaces. Overall the results were very similar. The most consistent trend was that the KEGG priorVAE model (without beta weighting) with a wild card dimension performed consistently highest

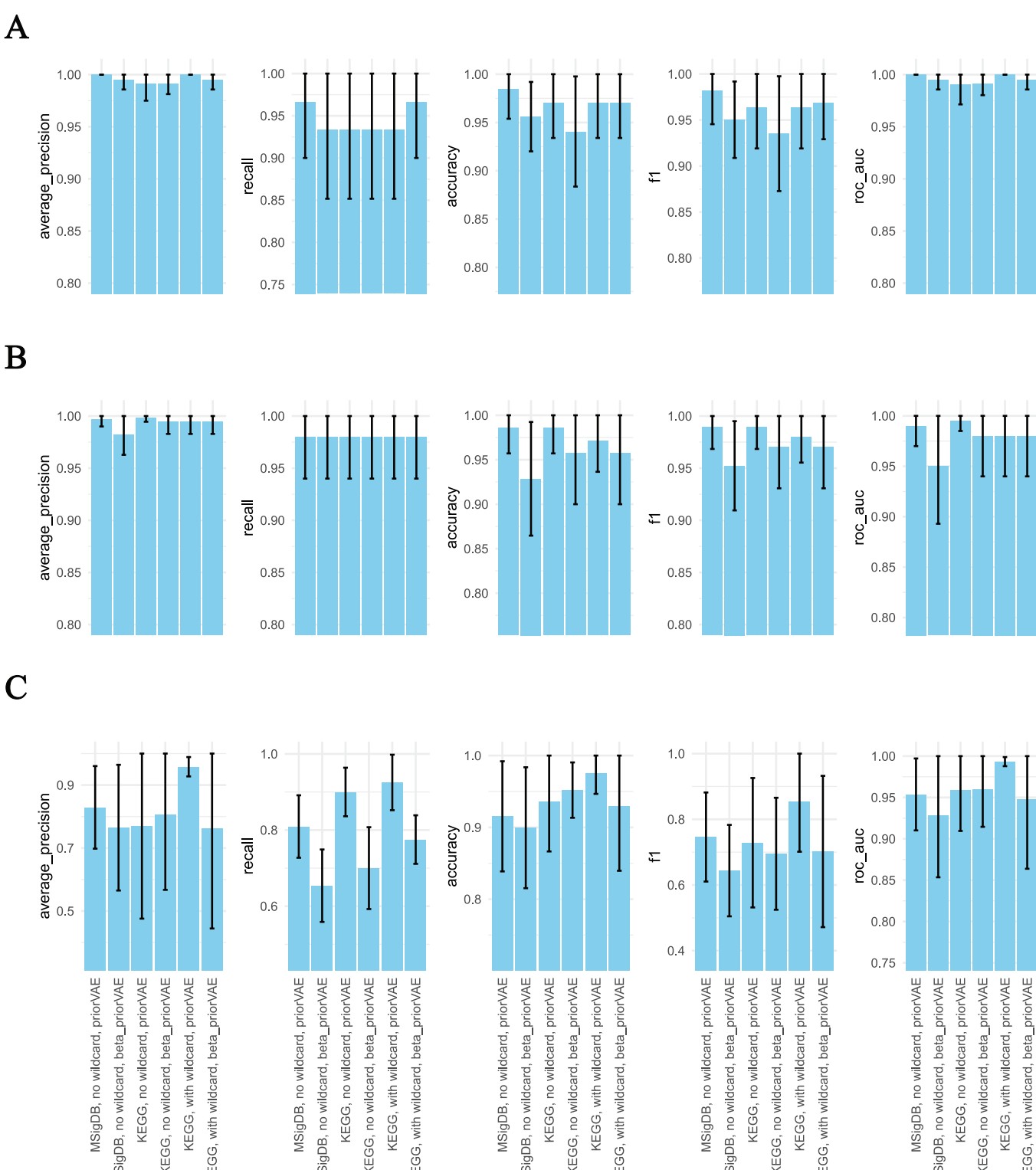

**Fig 4. The performance of the KEGG-based models across classification tasks.** KEGG-based models are compared to MSigDB-based models. Sample classification was based on multivariate logistic regression models as a function of the latent representation provided by each of the autoencoder architectures.

with regard to tissue comparison. Further results of using a KEGG-based latent space are reported later in the section on differential analysis.

## Influence of priors on learned latent representation

The goal of bringing pathway-derived priors into the model is to produce latent representations that both accurately represent the full transcriptome and also directly correspond to recognizable biological concepts. However, these goals are at odds, as we reported in (Fig 5 and S6 Fig). In the case of the priorVAE, the latent variable *allograft rejection* does retain a high correlation with the prior scores (R = 0.82). However, all the other 49 pathways are with a correlation weaker than 0.5, and only six pathways are to some extent correlated with the prior, even with a far less stringent threshold of 0.4. 21 out of 50 pathways correlate negatively with the prior scores. In contrast, the latent variables of the beta-priorVAE retain their connection to their pathways, with 48 of 50 pathways correlating higher than 0.6. Fig 5C indicates that the beta-priorVAE provides latent values with a high level of semantic meaning, providing a direct means for interpreting complex transcriptome datasets.

## Differential analysis

The beta-priorVAE provides a simplified representation of the transcriptome in terms of 50 features corresponding to 50 pathways in the case of MSigDB, and 149 for KEGG. A comparative analysis across samples and conditions is now possible directly on the basis of these features. Analogous to performing a differential expression analysis on genes, we performed a differential latent variable (i.e. differential pathway) analysis using the model results. We included the four disease scenarios from the classification benchmark in this differential analysis: (i) lung adenocarcinoma vs small cell lung cancer (ii) lung vs breast adenocarcinoma, (iii) lung adenocarcinoma vs healthy lung, and (iv) leukemia vs normal bone marrow.

Applying the MSigDB autoencoder to the differential analysis of adenocarcinoma vs small cell lung cancer, identified *complement* and *TNF-alpha signaling via NFKB* as major distinguishing factors (Fig 6). The KEGG model was in agreement regarding this latter pathway and also proposed *IL-17 Signalling*. In the case of leukemia versus healthy bone marrow, the top MSigDB pathway identified by the beta-priorVAE was *UV response* followed by *epithelial mesenchymal transition* [44, 45] (S7 Fig). The KEGG model proposed different differential pathways, including *ECM-receptor interaction* and *mRNA surveillance*. When comparing lung adenocarcinoma vs healthy, the top MSigDB pathway, *angiogenesis* was extremely significant ($p < 10^{-25}$), followed by proliferation-related pathways (S8 Fig). The KEGG model produced *vitamin B6 metabolism* and again *ECM-receptor interaction*, as was the case for leukemia. In the case of breast vs lung adenocarcinoma, the MSigDB pathway identified coagulation as the top hit, followed by *xenobiotic metabolism* (S9 Fig). We then applied a more traditional bioinformatic method of analysis as a point of comparison with the differential pathway analysis reported above. Specifically, we performed differential gene expression analysis, followed by gene set enrichment analysis, on the adenocarcinoma vs small cell lung cancer datasets. The resulting pathway analysis for KEGG is reported in Table 3. This approach resulted in several top hits consistent with RNA degradation artifacts. However, because the analysis included the full set of KEGG pathways (recall that the KEGG pathways for the latent spaces were filtered to exclude diseases), on if the gene sets found were in fact small cell lung carcinoma. The combination of these results indicates that the differential gene analysis is valid, but suffers from the data quality of the samples.

A final analysis was performed due to the observation that the comparison between the priorVAE and beta-priorVAE show clearly a trade-off between capturing the biological

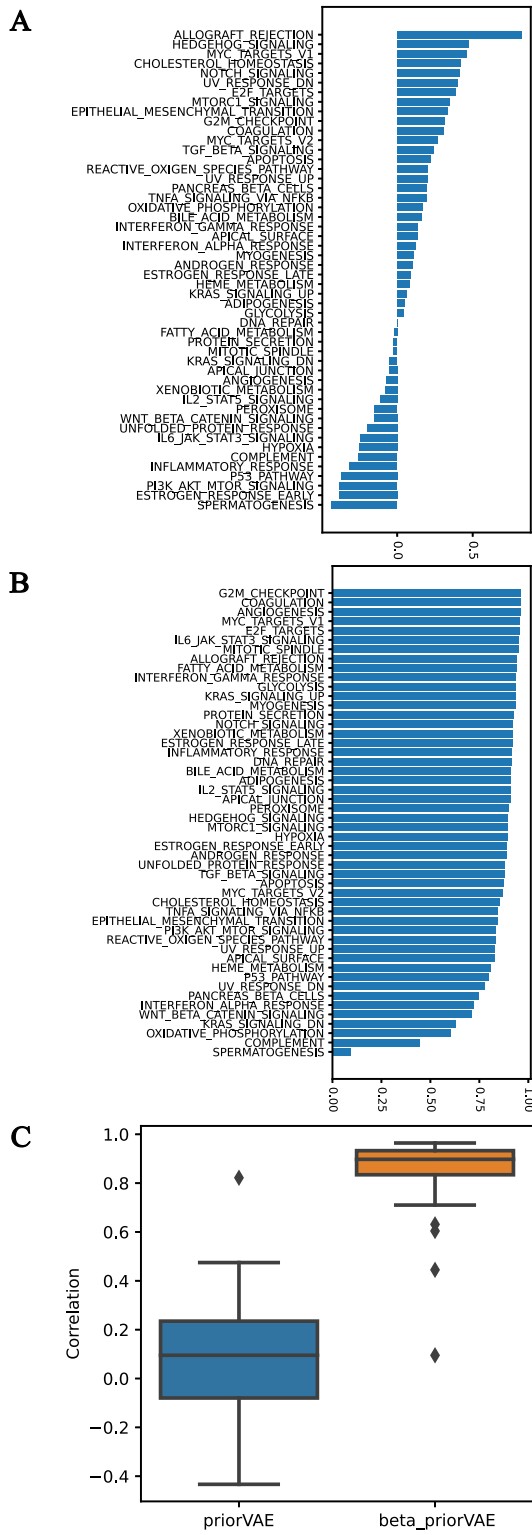

**Fig 5. The semantic meaningfulness of the latent variables in the prior-based models, shown as the correlation between the biological priors and the latent $\mu$ of prior-based models on the test set.** The correlation of each dimension is shown in A: for priorVAE and B: for beta-priorVAE. Subplot C summarizes these correlations to directly compare the semantic interpretability of the two models.

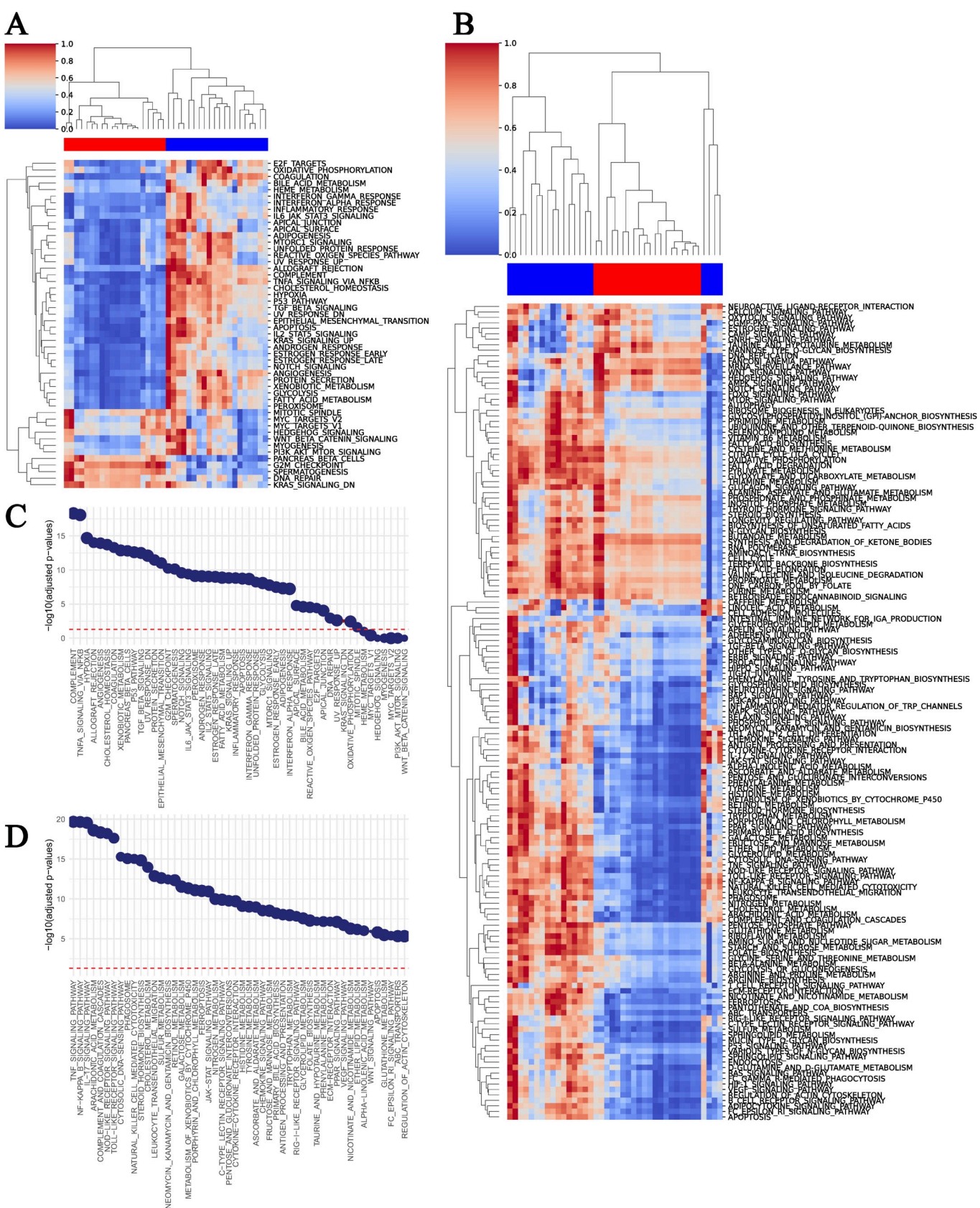

**Fig 6. Differential latent space analysis of adenocarcinoma vs small cell lung cancer.** Heatmaps show latent values pathways defined by A: MSigDB and B: KEGG. C and D show the top differentially expressed latent variables based on the p-values for MSigDB and KEGG respectively.

**Table 3. Overrepresented KEGG gene sets by traditional differential expression analysis.**

| KEGG Term | P.value | Adjusted.P.value | Combined.Score |
|---|---|---|---|
| Ribosome | $1.93 * 10^{-7}$ | $5.47 * 10^{-5}$ | 37.43 |
| Protein processing in endoplasmic reticulum | $3.58 * 10^{-7}$ | $5.47 * 10^{-5}$ | 34.22 |
| Lysosome | $2.12 * 10^{-6}$ | $2.16 * 10^{-4}$ | 31.95 |
| Endocytosis | $4.07 * 10^{-6}$ | $3.10 * 10^{-4}$ | 23.38 |
| Mitophagy | $7.17 * 10^{-6}$ | $4.37 * 10^{-4}$ | 37.22 |
| Autophagy | $4.30 * 10^{-5}$ | $2.19 * 10^{-3}$ | 21.51 |
| Hedgehog signaling pathway | $6.37 * 10^{-4}$ | $2.78 * 10^{-2}$ | 20.44 |
| Glycosaminoglycan degradation | $9.26 * 10^{-4}$ | $3.14 * 10^{-2}$ | 33.00 |
| Valine, leucine and isoleucine degradation | $8.94 * 10^{-4}$ | $3.14 * 10^{-2}$ | 18.78 |
| Non-small cell lung cancer | $1.56 * 10^{-3}$ | $4.76 * 10^{-2}$ | 14.45 |

variability in the models' latent space, but meanwhile, adhering to prior biological concepts found in the set of Hallmark pathways. To further investigate the effect of hyperparameter beta on performance, we ran the benchmarks across a range of values for beta (S10 Fig) The classification performance seems to decrease consistently with an increasing beta, although for beta values up to 100, the trend is close to flat. This implies that we can find a beta value that balances the need to capture biological features and, at the same time, adhere to the pathway labels provided via the priors.

## Discussion

The results of these experiments demonstrate that autoencoders are capable of generating a simplified representation of a transcriptome that still retains the key biological information necessary to differentiate different cells under different conditions. Furthermore, it is also possible to constrain the training process in a way that forces the network to find a latent representation corresponding to human-understandable biological concepts. Here we have achieved this by taking advantage of the VAE framework, which allows for integrating prior knowledge. There is a trade-off between efficiently representing the complexity of a transcriptome, and adhering to a panel of chosen biological concepts, in our case, defined by 50 Hallmark pathways.

An early decision made in this pipeline was to proceed with gene-level instead of transcript-level or community-level input. The motivation for reducing the size of the input to the neural network is that oversized inputs increase the risk of overfitting and reduce the likelihood that the model will find a latent representation that is a good abstraction of the original data. In our hands, the community detection-based approach, which would have reduced the input space the most did not perform better than the larger gene-level input, although it was better than using the raw transcript input. However, we cannot exclude the possibility that superior community detection approaches, perhaps sourced from network inference methods would provide yet a further improvement to our models.

The primary goal of utilizing pathway-based priors in the priorVAE and beta-priorVAE models was to generate a latent space that would be immediately interpretable to a biologist because the model will describe a transcriptome in terms of features that are familiar to a biologist. However, we have observed that latent features do not always retain the identity of their associated priors. In the case of the priorVAE (i.e. beta = 1), the model has substantial freedom to deviate from the pathway priors, and in fact, does so for many features. However, by

boosting the requirement to adhere to the pathway concepts with the beta hyperparameter, the immediate biological interpretation of the latent space is achievable (Fig 5).

Although beta does help direct the model to retain the association of the latent variables with their respective pathways, a side effect of this "strictness" is that the model no longer has the flexibility to learn sources of variation in the dataset that are not covered by the pathway definitions. For example, technical covariates such as RNA degradation could be such a source of variation. The results here show that in fact the introduction of wild card latent variables that are not contained to associate with any particular prior does improve the model accuracy. In particular, this was the case with regard to the tissue classification benchmark.

The beta-priorVAE with the current setting shows an overall satisfying correlation between the latent variables and prior scores, which enables an interpretation of the biological pathways involved in the chosen vignette. In the use case of adenocarcinoma vs small cell lung cancer, the t-test results indicated that TNF and NFKB signaling pathways were major distinguishing factors (Fig 6). Notably, there was also consensus between the KEGG and MSigDB models on this point. Although NFKB is generally implicated in cancer [46], it has been reported that its link with prognosis specific to adenocarcinoma [47]. Furthermore, *coagulation* is among the top 7 significant differentiators between the two lung cancer phenotypes in the MSigDB model. While coagulation function is associated with the prognosis in NSCLC patients as described in [48, 49], both [50] and [51] reported an absence of similar correlation between coagulation and the SCLC prognosis. In the KEGG model, *arachidonic acid metabolism* was a top hit, and this potentially also a differentiating factor between the cancer types because genes of this pathway are upregulated specifically in adenocarcinoma [52]. Thus, the literature reports that The literature is, therefore, consistent with the latent representation in that the concept of coagulation is a differentiating feature between the two diseases. Thus, the evaluation supports the feasibility of such architecture in making transcriptomes more intuitively transparent and interpretable.

Although our primary motivation for including priors in our VAE was to make the latent space directly interpretable, the traditional motivation for including priors was to increase model accuracy. For most of our benchmarks, model accuracy generally decreased when we increased the emphasis on priors via the beta parameter. This implies that the reconstruction portion of the loss function is the main driver of performance in these benchmarks in comparison to the KL divergence term. However, an interesting point of comparison in our experiments is between the beta-simpleVAE and beta-priorVAE, which have the same emphasis on priors (i.e., the same betas), but in the latter model, prior biological knowledge is incorporated. Table 2 and S10 Fig show an increase in performance when using a biological prior vs. a unit Gaussian prior, indicating that in this local comparison, prior biological information can be beneficial.

We have provided evidence that the key sources of biological variation are captured in the latent space. At the primary level, the successful reconstruction as demonstrated by the reconstruction loss, as well as high correlation coefficients between inputs and outputs, indicate that the latent representations are reliable. This is further supported by the fact that cancer types and tissue types can be distinguished using only the latent features. However, the performance for classifying tissues is surprisingly mediocre. The question is whether this is a limitation of the information found in the latent representation or an inadequate classification procedure. The latter scenario is supported by the high reconstruction accuracy and the fact that the multivariate logistic regression may be inadequate without proper feature selection.

## Conclusion

The training and post hoc experiments demonstrate that (i) autoencoder models can find simplified representations of transcriptomes that still retain biological information, (ii) using pathway-derived priors, we can encourage the models to find latent representations that still adhere to concepts that are familiar to biologists, and (iii) latent features can provide a direct means of comparison among samples and conditions that can provide an immediate biological interpretation. This area of research should be explored further, with attention to alternate pathway definitions to define the priors and thus the latent space, additional model architectures, and integration into bioinformatic workflows.

## Supporting information

**S1 Fig. Input vs output correlations for various input types: Transcript, gene, or community input.** The pairwise correlation plots include both input and output transcriptomes. The Boxplots depict the correlations between every input and their respective output.
(PDF)

**S2 Fig. Heatmap reporting the correlation between input and output transcriptomes annotated by tissue type.** The rows and columns are ordered by sample, so the diagonal reports the correlation for the input-output pair for one sample.
(PDF)

**S3 Fig. Analysis of reconstruction performance at the gene level.** The absolute error between input and output gene levels was calculated for each gene and plotted as a function of the average expression level.
(PDF)

**S4 Fig. t-SNE plots comparing original input and reconstructions across tissue types.** A Shows the performance for the prior VAE and B for beta-prior VAE.
(PDF)

**S5 Fig. Post-hoc hyperparameter analysis.** A sweep (random search) of the hyperparameter space was performed, resulting in 60 evaluated models. A: importance and correlation metrics provided by wandb. B: model results summarized by encoder architecture. X-axis labels refer to the size of the hidden layers in the encoder and decoder. C: Sweep results for the beta-priorVAE model across beta values, reporting the correlation between latent values and priors. D: Sweep results for beta, reconstruction loss, KL loss, and total loss. E-G: Effects of beta sweep on classification tasks for E: leukemia vs health, F: lung vs breast cancer, and G: tissue.
(PDF)

**S6 Fig. The scatter plot of the priors (x-axis) and the latent $\mu$ (y-axis) of all test samples for A: priorVAE and B: beta-priorVAE.** These plots show the extent to which the model adheres to the original meaning of the pathway labels.
(ZIP)

**S7 Fig. Differential latent space analysis of adenocarcinoma vs non-small cell lung cancer.** Heatmaps show latent values pathways defined by A: MSigDB and B: KEGG. C and D show the top differentially expressed latent variables based on the p-value for MSigDB and KEGG respectively.
(PDF)

**S8 Fig. Differential latent space analysis of adenocarcinoma vs healthy lung samples.** Heatmaps show latent values pathways defined by A: MSigDB and B: KEGG. C and D show the top differentially expressed latent variables based on the p-value for MSigDB and KEGG respectively.
(PDF)

**S9 Fig. Differential latent space analysis of breast vs lung adenocarcinoma.** Heatmaps show latent values pathways defined by A: MSigDB and B: KEGG. C and D show the top differentially expressed latent variables based on the p-value for MSigDB and KEGG respectively.
(PDF)

**S10 Fig. The average precision score of the beta-simpleVAE and the beta-priorVAE models on the tissue classification with different values for hyperparameter beta.**
(PDF)

## Author Contributions

**Conceptualization:** David S. DeLuca.

**Data curation:** Bin Liu.

**Formal analysis:** Bin Liu.

**Funding acquisition:** Thomas Illig, David S. DeLuca.

**Methodology:** Bodo Rosenhahn, David S. DeLuca.

**Project administration:** David S. DeLuca.

**Resources:** David S. DeLuca.

**Software:** Bin Liu.

**Supervision:** Bodo Rosenhahn, Thomas Illig, David S. DeLuca.

**Writing – original draft:** Bin Liu, David S. DeLuca.

**Writing – review & editing:** Bin Liu, David S. DeLuca.

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
