## [Decision Letter · Decision Letter 0]

5 Aug 2023

Dear Dr. DeLuca,

Thank you very much for submitting your manuscript "A variational autoencoder trained with priors from canonical pathways increases the interpretability of transcriptome data" for consideration at PLOS Computational Biology.

As with all papers reviewed by the journal, your manuscript was reviewed by members of the editorial board and by several independent reviewers. In light of the reviews (below this email), we would like to invite the resubmission of a significantly-revised version that takes into account the reviewers' comments.

We would like you to respond to the specific feedback on major limitations that should be addressed before resubmission. The most detailed comments (from reviewer 3) include issues regarding the validation and benchmarking of performance, as well as issues of pathway representation from the selected pathways used in the development of the VAE. These issues should be addressed directly. Reviewers #1 and #2 raise other significant issues that need to be addressed. 

We cannot make any decision about publication until we have seen the revised manuscript and your response to the reviewers' comments. Your revised manuscript is also likely to be sent to reviewers for further evaluation.

Sincerely,

Greg Tucker-Kellogg, PhD

Guest Editor

PLOS Computational Biology

Mark Alber

Section Editor

PLOS Computational Biology

Reviewer's Responses to Questions

**Comments to the Authors:**

Reviewer #1: Here, the authors propose the use of autoencoder variations to increase interpretability of deep learning in transcriptomics data. More specifically, the novelty of this paper is the addition of canonical biological pathways as priors to the beta-variational autoencoder model. The authors reasoned that the ability to tune the prior weights with the beta hyperparameter ties the latent space to the curated biological pathways and henceforth providing better biological interpretability.

While the conceptual premise seems interesting, there are a few aspects I would like to see further explored:

- Table 2 provided showed that the introduction of beta drastically increases the reconstruction loss and reduces the KL divergence. However, the relationship between the beta hyperparameter and the two losses warrants more investigation – for example, how does the beta hyperparameter scale with respect to each loss? This may allow for a more fine-tuned approach to selecting the optimal amount of trade-off mentioned in the paper.

- The authors should expand on the classification metrics as well provide statistical tests to these metrics. Even though the authors attempted to justify the worse model accuracies with the poorer reconstruction losses, it remains unclear why the beta-prior models are underperforming compared to the simple counterparts. This argues against the premise and warrants more investigation. Providing more classification metrics may provide an insight into why the models are worse off. This also ties into previous point, if the reconstruction loss scales accordingly to beta hyperparameter, it will be evident in the classification metrics.

- Even though the argument for have biologically-relevant priors is logically sound, it requires more testing to determine if the effect is true. A good sanity check would be to introduce random biological sets and see if it reduces the model’s classification performance. Another potential check would be to see if there are specific genes that are heavily influencing the models. For example, by only using subsets of the gene sets, one can test for the robustness of the biological priors.

Other comments:

- While I agree that gene-level features are sufficient, I would like to see the effect of different normalization methods on the gene-level features. It is well known that different normalization methods can affect the downstream RNA-seq data analysis. As such, the authors should consider different normalization approaches.

- While the premise of the paper is to prove the utility of biologically-informed priors, it might be useful to see if different pathway definitions from different databases may result in different results.

- The scope of disease classification tasks should be expanded to include other types of cancers and disease to test the statistical robustness of the models. Different physiological conditions manifest into different statistical properties of the data distributions observed. As such, it will be prudent to find out what type of statistical distribution causes the model to perform better or worse.

- A good sanity check would be to see if the same pathways are derived when doing conventional differential gene expression analysis with the same classification groups. This may provide an insight into the advantages of using a machine-learning approach shown in this paper.

Reviewer #2: The authors proposed autoencoders to compute latent representation for transcriptome signals. Five autoencorders, such as simpleAE, simpleVAE, Beta-SimpleVAE, PriorVAE, and Beta-PriorVAE, were compared for the assessment. The problem in the manuscript is interesting, but there are several major concerns before the publication.

- The approaches need to be clarified. The data processing and pipeline for the transcript level input, gene level input, and community level input are not clear. I strongly encourage the authors to improve the method.

- What's definition of community level input? Why is important? And for the community detection, why is KNN-based graph the best solution? Why not network inference methods?

- Autoencoder is trained on each pathway or with whole gene level?

- What is the justification to set the node number and layer number determined?

- I am wondering if there are very similar works published before to compare the performance directly, rather than simple comparison with simple autoencoders.

Reviewer #3: Summary

This study develops an innovative approach to identifying biological processes undergoing perturbation from transcriptome data using variational autoencoders. The study achieves this through the incorporation of biological priors to direct the VAE networks to learn representations of transcriptomes that are based on biological concepts and could in principle be easier to interpret biologically. The authors show that both a simple fully-connected VAE and the novel prior-informed VAE can learn reduced representations of transcriptomes from high dimensions to 50 latent dimensions. These representations retain meaningful biological information that enable accurate reconstruction. While the fully connected VAE outperforms the prior-informed VAE in disease and organ classification tasks, it lacks directly interepretable latent dimensions. The prior-informed (beta-weighted) VAE not only solves the benchmark tasks but also provides semantically accurate latent features that map to biological pathways.

Overall, the work presented is innovative conceptually and methodologically. The main drawbacks include lack of rigor in benchmarking performance of the autoencoders as only a single metric (precision) is used to compare the models without consideration of the tradeoff to recall. There is need to assess the biological properties of the reconstructed transcriptomes related to the input datasets, especially an understanding of any relationships between reconstruction error and transcript levels of specific genes, pathways or tissue source/ disease state of the input samples. The ability of the prior-informed VAE to identify biological pathways that discriminate tissue source or disease state should also be compared to pathways that would be identified using conventional differential gene expression analysis and pathway enrichment. The validation of the prior-informed VAE's ability to enhance interpretability is weakened by the use of a set of 50-pathways as the source of priors for the latent space as the pathways are not comprehensive or representative of wide range of biological processes.

Strengths

-The approach to leverage VAE to identify differentially expressed biological pathways is innovative

-The approach addresses some key limitations of conventional approaches for differential gene expression analysis such as GSEA that require lists of well defined differentially expressed genes

-Unlike previous differential gene expression analysis approaches such as limma, DESeq2 and Seurat, the use of the prior-informed VAE does not assume linearity across samples

-The approach developed using autoencoders to enhance interpretability is distinct from other approaches that have been developed in the field using VAE. Unlike prior approaches that correlate phenotypes to latent features post-hoc or constrain the networks using known gene-pathway associations, the prior-informed VAE in this study enables the networks to leverage biological priors while still giving them freedom to learn relationships among genes

Major Limitations

-Authors need to perform a more detailed analysis of the samples where the correlation between the a given input transcriptome was not the most correlated to its reconstructed output to understand underlying mechanisms. This is to ensure future user can understand contexts in which the reconstruction may become unreliable

-There is need to assess any biases in reconstruction error at the gene and pathway level. For example, are there some genes or pathways whose transcript levels are more prone to higher error in reconstruction?

-There is need to assess reconstruction performance based on highly vs lowly expressed genes as biases in these gene groups could impact biological outcomes typically based on gene expression

-The tissue source/ treatment of the inputs need to be fully described in order to have a better understanding of the biological relevance of the results. Labeling the sample names with intuitive names will help in determining whether the observed correlations across tissues also capture tissue relationships

-A comparison of the classification performance using differential gene expression or expression profiles of each sample directly would also be informative and useful to the field to show any advantages and disadvantages

-Assessment of the performance needs to include recall and area under precision recall and the associated AUPRC curves. Without knowledge of the recall in relation to the precision, the utility of the high precision scores is low (including in the case where precision was 100%)

-The results of differential analysis features across conditions needs to be compared to conventional approaches e.g. GSEA. For example, when comparing adenocarcinoma vs. health tissue, what are the differentially expressed pathways obtained using typical methods (e.g. GSEA)?

-The Pathways captured by the 50-dimensions are not comprehensive so it seems to be a big biological flaw to try to reduce the transcriptome dimensionality to these 50 pathways and use those results for interpretability. It would be better to use more comprehensive ontologies e.g. GO or pathways (KEGG) or to modify the prior informed VAE to include a dimension for "Unknown pathways"

Minor Limitations

-It would be useful to show how well the network based clustering that was performed to identify representative genes is representative of the full transcriptome

-It would be insightful if the heatmaps of the inputs and outputs were visualized separately in addition to as shown in Fig. 2. Separate visualization figure for inputs vs. outputs would help to demonstrate whether tissue correlations based on reconstructed data recapitulate the tissue correlations based on inputs. This could also be visualized using tSNE

-Supplementary Fig. 1 is missing heatmaps for simple VAE and beta-simple VAE transcript associations

**Have the authors made all data and (if applicable) computational code underlying the findings in their manuscript fully available?**

Reviewer #1: Yes

Reviewer #2: Yes

Reviewer #3: **No: **There is a link to github repository but the page is not functional

PLOS authors have the option to publish the peer review history of their article (what does this mean?). If published, this will include your full peer review and any attached files.

Reviewer #1: No

Reviewer #2: No

Reviewer #3: **Yes: **Geoffrey H. Siwo

Figure Files:

Data Requirements:

Please n

---

## [Decision Letter · Decision Letter 1]

11 Jun 2024

Dear Dr. DeLuca,

We are pleased to inform you that your manuscript 'A variational autoencoder trained with priors from canonical pathways increases the interpretability of transcriptome data' has been provisionally accepted for publication in PLOS Computational Biology.

Best regards,

Mark Alber, Ph.D.

Section Editor

PLOS Computational Biology

Mark Alber

Section Editor

PLOS Computational Biology

Reviewer's Responses to Questions

**Comments to the Authors:**

Reviewer #1: I am satisfied with the new revision.

**Have the authors made all data and (if applicable) computational code underlying the findings in their manuscript fully available?**

Reviewer #1: Yes

PLOS authors have the option to publish the peer review history of their article (what does this mean?). If published, this will include your full peer review and any attached files.

Reviewer #1: No

---

## [Editor Report · Acceptance letter]

26 Jun 2024

PCOMPBIOL-D-23-00803R1 

A variational autoencoder trained with priors from canonical pathways increases the interpretability of transcriptome data

Dear Dr DeLuca,

I am pleased to inform you that your manuscript has been formally accepted for publication in PLOS Computational Biology. Your manuscript is now with our production department and you will be notified of the publication date in due course.

With kind regards,

Livia Horvath
